# Role of JAK/STAT3 Signaling in the Regulation of Metastasis, the Transition of Cancer Stem Cells, and Chemoresistance of Cancer by Epithelial–Mesenchymal Transition

**DOI:** 10.3390/cells9010217

**Published:** 2020-01-15

**Authors:** Wook Jin

**Affiliations:** Laboratory of Molecular Disease and Cell Regulation, Department of Biochemistry, School of Medicine, Gachon University, Incheon 21999, Korea; jinwo@gachon.ac.kr

**Keywords:** JAK/STAT, JAK/STAT, metastasis, the transition of cancer stem cells (CSCs), chemoresistance, epithelial–mesenchymal transition (EMT)

## Abstract

The JAK/STAT3 signaling pathway plays an essential role in various types of cancers. Activation of this pathway leads to increased tumorigenic and metastatic ability, the transition of cancer stem cells (CSCs), and chemoresistance in cancer via enhancing the epithelial–mesenchymal transition (EMT). EMT acts as a critical regulator in the progression of cancer and is involved in regulating invasion, spread, and survival. Furthermore, accumulating evidence indicates the failure of conventional therapies due to the acquisition of CSC properties. In this review, we summarize the effects of JAK/STAT3 activation on EMT and the generation of CSCs. Moreover, we discuss cutting-edge data on the link between EMT and CSCs in the tumor microenvironment that involves a previously unknown function of miRNAs, and also discuss new regulators of the JAK/STAT3 signaling pathway.

## 1. Introduction

The Janus kinase (JAK)/signal transducer and activator of transcription (STAT) pathway played a crucial role in many biological functions during the multistep development of human tumors, including proliferation, inflammation, and survival. The JAK/STAT signaling pathway comprises of the receptor and adaptor proteins of interleukin 6 (IL-6), interferon-alpha (IFN-α), and interferon-gamma (IFN-γ) that mediate pleiotropic functions upon binding to their respective ligands [1,2].

The IL-6 family of cytokine comprises IL-6, IL-11, IL-27, IL-31, oncostatin M (OSM), cardiotrophin 1 (CT-1), ciliary neurotrophic factor (CNTF), cardiotrophin-like cytokine factor 1 (CLCF1), and leukemia inhibitory factor (LIF). Elevated expression of the cytokines belonging to this family is implicated in the development of various human diseases [3,4]. Upon binding IL-6, the IL-6 receptor-α (IL-6R) forms a complex with glycoprotein 130 (IL-6Rβ), and subsequently, triggers the activation of receptor-associated JAK1, JAK2, and tyrosine-protein kinase 2 (TYK2) pathways [4,5].

There are four JAK family non-receptor tyrosine kinases, JAK1, JAK2, JAK3, and TYK2. JAK1, JAK2, and TYK2 are ubiquitously expressed, whereas JAK3 is predominantly expressed in hematopoietic cells [6]. The JAK family is characterized by the presence of four unique domains, four-point-one, ezrin, radixin, moesin (FERM); Src homology 2 (SH2); pseudokinase; and kinase domains. The FERM and SH2 domains facilitate association with cytokine receptors and regulate the catalytic activity [7]. The pseudokinase domain, which interacts with the kinase domain, acts as a suppressor of the kinase domain’s catalytic activity and subsequently activates STAT1, 3, and 5 [8].

Until now, seven members of the STAT family (STATs 1–4, 5α, 5β, and 6) have been identified. Each of the STAT proteins shares highly conserved domains, including amino-terminal, coiled-coil, DNA binding, SH2, and transactivation domains [9]. The Asp170 residue in the helix α1 of the coiled-coil domain of STAT3 interacts with other transcription factors [10], and tyrosine phosphorylation of STAT3 by IL-6 is required for its receptor binding, dimerization, nuclear translocation, and DNA binding [11]. The SH2 domain is essential for STAT-cytokine receptor interactions as it recognizes the tyrosine residues in the cytokine receptors and forms stable homo- or heterodimers with other STAT proteins [12,13]. Cytokines induce the dimerization of STAT3 through the acetylation of Lys685 in the SH2 domain of STAT3, which is associated with the histone acetyltransferase p300 [14]. Besides, the N-terminal domain of STAT3 has multiple functions, including STAT3 tetramer stabilization, cooperative DNA binding, nuclear translocation, and protein–protein interactions [15] (Figure 1).

## 2. Role of IL-6/JAK/STAT3 in the Induction of EMT

STAT proteins are differentially implicated in cancer tumorigenesis. Although STAT1 is known to be involved in mediating the anti-tumor immunity and other STAT families are known to be involved in the promotion of cancer development, it is STAT3 that is most well studied as a significant intrinsic transcription factor in the induction of the EMT and in the pathogenesis of cancer (Figure 2) [16]. IL-6/JAK2/STAT3 activation enhances metastasis via induction of EMT by the upregulation of EMT-inducing transcription factors (EMT-TFs; Snail, Zeb1, JUNB, and Twist-1) and increases cell motility via focal adhesion kinase (FAK) activation [17,18,19,20]. In prostate cancer, paracrine IL-6/JAK2/STAT3 stimulates the autocrine IL-6 loop, and IGF-IR activation induced by both IL-6 and IGF enhances EMT through induction of the STAT3/NANOG/Slug axis [21,22]. 

STAT3 acts as a transcriptional activator through binding to the promoter of target genes in a tyrosine phosphorylation-dependent manner [23]. For instance, once activated, STAT3, in combination with estrogen receptor (ER), binds to the LIV-1 promoter leading to the induction of LIV-1 expression, followed by proteolytic N-terminal cleavage of the LIV-1 protein and its subsequent translocation to the plasma membrane. Subsequently, LIV-1 stabilizes Snail through inactivation of GSK3β [24,25,26]. In addition, RANKL is known to enhance EMT in prostate cancer through the STAT3/LIV-1 axis [27]. Moreover, IL-8-mediated production of IL-6 and TGF-β1 promotes binding of STAT3 to AUF-1 promoter, leading to the activation of breast stromal fibroblasts through the reduction of p16, p21, and p53 levels in a paracrine manner. In turn, elevated AUF-1 levels lead to enhanced EMT due to increased expression of SDF-1, α-SMA, TGF-β1, and IL-6 that results from binding of AUF1 to their respective promoters [28,29]. Furthermore, increased expression of Fra-1 [30] and PTTG1 [31] due to the binding of STAT3 to their promoters induce invasion and facilitates resistance to androgen deprivation therapy (AR). Additionally, induced RTVP-1 expression via binding of both C/EBPβ and STAT3 to the RTVP-1 promoter results in maintenance of the stemness property of glioblastoma cells and is associated with poor clinical outcomes in patients [32]. Besides, STAT3 induces UHRF1 expression via direct binding to its promoter. This is especially important in the context of cancer biology as the PHD and SRA domains of UHRF1 are involved in silencing the tumor suppressor genes in CRC cells [33]. The hallmarks of cancer coordinated by the STAT3-dependent transcriptional regulation of target proteins directly underpin the cellular responses.

## 3. Critical Modulators of the JAK2/STAT3 Signaling Pathway in EMT

The activation of JAK2/STAT3 signaling is triggered by various hormones, cytokines (including the IL-6 family), and growth factors through a variety of molecular mechanisms. OSM, another IL-6 cytokine family member, is crucial for JAK/STAT activation. It interacts with the extracellular matrix (ECM) proteins, and OSM/OSM receptor (OSMR)-mediated JAK2/STAT3/FAK/Src activation promotes EMT and CSC production [34,35,36,37,38,39]. Other pro-inflammatory cytokines, IL-32β and IL-23, are highly expressed in cancers. STAT3 activation by IL-23/IL23R and via IL-32β-induced VEGF leads to increased migration and invasion of breast cancer (BC) and gastric cancer (GC) cells [40,41]. Additionally, the FAM3 family of created cytokines, ILEI-induced PDGF-C/PDGF-Rβ [42], or PDGF-Rα activation [43], enhance EMT and aggressiveness by increasing p-STAT3 via Src activation. Repression of hnRNP E1 by non-canonical TGF-β-induced Akt2 increases ILEI expression, and ILEI/ILFR promotes EMT and CSC generation [44,45].

Hormones, growth factors, and chemokines are associated with JAK/STAT3 activation. The gastrointestinal (GI) peptide hormone, gastrin, promotes the invasiveness of cancer cells by the induction of MMP-1, -2, and -9 secretion [46]. Gastrin-mediated activation of JAK2/STAT3 and PI3K/AKT pathways inhibit cell–cell adhesion and cell motility [47]. Furthermore, gastrin and its receptor, GRPR, promote EMT by increasing Snail and reducing E-cadherin expression [9]. However, polyclonal antibody stimulator (PAS), an anti-gastrin cancer vaccine, inhibits tumor formation, metastases of pancreatic cancer, and gastrin-induced EMT [48]. CXCL12-induced CXCR7 receptor results in increased IL-8 and VEGF levels, and subsequently, EMT by induction of Snail through the activation of AKT, ERK, and STAT3 in bladder cancer and BC [49,50].

The heparin-binding growth factor, Midkine (MK), induces EMT via activating the nuclear accumulation of the Notch-2 receptor through the formation of the MK-Notch-22 complex [51]. In addition, after the formation of the MK-high affinity receptor, this complex is associated with JAK1/JAK2 and subsequently activates JAK/STAT signaling [52]. Moreover, MK engenders a cross-talk between the Notch signaling and JAK2/STAT3 signaling by inducing an interaction between the JAK2/STAT3 complex and Hes1, which is the downstream target of Notch-2 [51,53]. Recent studies have demonstrated that induced MK is significantly associated with chemoresistance in pancreatic cancer via the activation of NF-κB signaling and Hes1-induced JAK2/STAT3 signaling through cleavage and activation of MK-mediated Notch signaling [54,55]. Intriguingly, the binding of estradiol (E2) to ERβ results in the activated ER-β binding to the estrogen response element within the MK promoter, thereby leading to the induction of MK transcription, which enhances the EMT [56]. Hepatocyte growth factor (HGF), its splice variant, and its receptor (c-MET) contribute to the activation of STAT3 and its downstream cascades. Following HGF binding, the docking motif within the C-terminal end of activated c-MET associates with the SH2 domain of STAT3 and phosphorylates STAT3 [57,58]. However, suppressor of cytokine signaling (SOCS)-1 reduces STAT3 activation by proteasomal degradation of c-MET through direct interaction with c-Met [59,60].

Non-canonical Wnt signaling, which is independent of the β-catenin/TCF pathway, induces EMT through STAT3 activation. Upregulated Wnt5a/b and its receptor, Wnt receptor, Frizzled2 (Fzd2), induce cell migration by non-canonical pathways, and activation of STAT3 by the Fzd2/Fyn/STAT3 complex upregulates STAT3 target genes [61].

G-protein coupled receptors (GPCRs) and human GH (hGH) have been reported to be involved in the regulation of JAK/STAT signaling in cancer. Activation of PAF/platelet-activating factor receptor (PAFR) promotes lung cancer progression via the activation of Src/STAT3 or JAK2/STAT3 [62]. Moreover, hGH, which acts as a central endocrine regulator, enhances the oncogenicity of endometrial carcinoma via the activation of STAT3 in a JAK2/src-dependent manner [63]. Critical modulator-mediated JAK/STAT signaling showed a mechanistic link to EMT, invasiveness, malignancy, recurrence, anticancer drug resistance, and generation of cancer stem cells (Figure 1A).

## 4. Orchestrators of JAK2/STAT3 Activation in EMT

The contribution of intracellular signaling components to the induction of EMT through the activation of the JAK/STAT signaling in cancer progression has been extensively studied (Figure 1B). 

### 4.1. Tyrosine and Serine/Threonine Kinases as Orchestrators

Potential mechanisms by which tyrosine or serine/threonine kinases activate the JAK/STAT3 signaling. Nuclear interactions, as well as cytoplasmic interactions between EGFR and STAT3, increase the expression of iNOS, cyclin D1, and c-fos via direct binding of EGFR/STAT3 complex to their promoters [64]. Moreover, aberrant EGF/EGFR signaling enhances EMT and cisplatin-resistance through the activation of JAK2/STAT3 via the following: increasing IL-6 and LIF production [65], src-induced Zeb1 and Zeb2 upregulation [66], and increasing Twist-1 expression via binding of STAT3 to the promoter of Twist-1 [67,68]. PTK6 interacts with the EGFR family—including EGFR, HER2, HER3—and aggravates cancer through activation of RAS/MAPK, PI3K/AKT, and STAT3 [69]. Besides, IFN-α induces apoptosis by activating JNK and p38 MAPK signaling. Additionally, IFN-α induces apoptosis via promotion of caspase 3-mediated degradation of gelsolin and inhibiting STAT3 activity. However, EGF/EGFR signaling inhibits IFN-α-mediated apoptosis via activation of Ras/ERK signaling and inhibition of caspase-3 activation [70,71,72]. In addition, Fyn-, EFG-, and TGFβ1-mediated PYK2 activation are involved in the progression and survival of BC patients via activation of STAT3, and subsequently, PYK2-induced STAT3 increases PYK2 and c-MET via activation of the positive feedback loop [73,74]. Moreover, Src- and Lyn induces the activation of Sgk269 (PEAK1), a cytoplasmic pseudokinase that is highly expressed in a basal subtype of BC and promotes EMT through both STAT3 and ERK activation [75,76]. Another pseudokinase, Sgk223 (Pragmin), promotes invasion and EMT in pancreatic cancer via activation of JAK1/STAT3 by the formation of Sgk223-STAT3 complex [77]. Interestingly, neurotrophin receptors, TrkB and TrkC, are regulators of JAK/STAT3 signaling. Both TrkB and TrkC increase JAK2 stability through inhibition of SOCS-3-mediated JAK2 degradation via direct interaction with JAK2, thereby resulting in the induction of EMT and metastasis of BC [78,79]. Additionally, William’s syndrome transcription factor (WSTF), a tyrosine kinase, promotes tumor growth via activation of AKT and STAT3 in lung cancer [80]. 

Serine/threonine kinases as the components of the ion channel are involved in STAT3 activation. The production of calcium signal by EGF induces EMT via activation of STAT3 by increasing the levels of TRPM7, a serine/threonine kinase [81]. Another serine/threonine kinase, ILK, induces EMT by increasing the activation of JAK2/STAT3 [82] and upregulation of SMOC-2 [83]. Mesenchymal cells resistant to EGFR inhibitors showed increased AKT and STAT3 activation; this was mediated via elevated levels of ILK [84]. Moreover, ILK enhances EMT and angiogenesis through induction of the MUC1-C oncoprotein by IL6/STAT3 and by the inhibition of MUC1-C degradation through the reduction of PKCδ, which is involved in ubiquitin-dependent degradation of MUC1-C [85]. Subsequently, MUC1-C and activated STAT3 induce MUC1 transcription by binding to the MUC1 promoter [86]. Furthermore, TRPP2 and TRPC1, which are involved in enhancing Ca^2+^ permeability, enhance metastasis by induction of EMT through increased STAT3 expression [87]. TRPC1-induced AKT and STAT3 activation increase Snail levels through the upregulation of HIF-1α and LC3BII (a hypoxia-induced autophagy marker) [88]. Increased interleukin production (IL-1α and IL-8) due to the action of PIM-3 [89] and PIM-2 as other serine/threonine kinases promotes the metastasis of BC via STAT3 activation and activated STAT3 upregulates PIM-2 expression by forming a positive feedback loop [90]. Furthermore, IL-6/JAK/STAT3 activation induces c-Myc expression through the upregulation of PIM-1 to enhance EMT and stemness [91].

### 4.2. Other Proteins as Orchestrators

Several proteins regulate STAT3 activation through direct interaction with STAT3. Pin1 is highly expressed in a majority of cancers [92,93], and elevation in Pin1 levels by either OSM or IL-6 involves the activation of both STAT3 and NF-κB through direct binding with the pSer/Thr-Pro motifs of STAT3 and p65 in the nucleus that results in promotion of STAT3 transcriptional activation [94,95]. The activation of RAC1 (RAC1–GTP) by TGF-β-induced PKC-ι results in STAT3 activation via RAC1–GTP-mediated triggering of the IL-6/JAK and Rac1–JNK pathways, or via its interaction with STAT3 [96,97,98,99]. Additionally, the ATP-independent chaperone Hsp27 (HSPB1) interacts with STAT3 in response to cellular stress-mediated [100], and both HSPB1 and TMPRSS4, induce EMT through the activation of the IL-6/STAT3/Twist-1 axis [100,101].

Diverse proteins are upregulated during cancer progression and facilitate constitutive JAK/STAT activation. The levels of G3BP1, which plays a critical role in stress granule (SG) formation, are increased in renal cell carcinoma (RCC) and promote IL6/STAT3 activation [102]. FUZ is a critical regulator of cilia structure, whose expression is associated with poor prognosis in lung cancer patients; this effect of FUZ is mediated through the activation of ERK1/2 and JAK2/STAT3 signaling via the formation of the FUZ–BNIP3 complex [103]. Moreover, EIF5A2 increases STAT3 stabilization, which via binding to the TGF-β1 promoter, enhances TGF-β1 expression [104,105]. Furthermore, SMYD3 induces STAT3 activation by increasing JAK1 and JAK2 expression via binding to their promoter regions. Furthermore, SMYD3 is a component of the SMYD3/H3K4Me3/RNA pol-II complex that enhances the transcription of genes that are involved in cancer-related pathways [106].

The activation of fundamental components of cellular metabolism and DNA methylation involves tumor progression by regulation of STAT3 activity. Upregulation of G6PD, an enzyme involved in the pentose phosphate pathway, and PKM2, the key enzyme catalyzing the final step of glucose metabolism is correlated with metastasis and poor survival of patients with HCC and ESCC and promotion of EMT via STAT3 and STAT5 [107,108]. Moreover, a lysine methyltransferase, EZH2 as a part of the EZH2-containing PRC2 complex, maintain gene silencing through hypermethylation of tumor suppressors [109,110], and p-AKT-induced EZH enhances tumor progression through activation and methylation of STAT3 by direct association [111]. Moreover, IL-6/STAT3-induced DNMT1 enhances the metastatic potential and reduces radiosensitization in AR-prostate cancer [112] and accumulates myeloid-derived suppressor cells, all of which are responsible for the accelerated tumor growth [113]. 

The Annexin (Anx) family contains calcium-dependent phospholipid-binding proteins and is associated with the invasiveness of cancer. AnxA1 induces invasiveness, drug-resistance, and generation of CSCs in prostate and pancreatic cancer by increasing the p-STAT3 level [114]. In addition, EGF/EGFR and TGF-β induce AnxA2 activation, following which p-AnxA2 directly interacts with STAT3 and subsequently enhances EMT in BC and CRC [115,116].

Hypoxia-responsive proteins are involved in STAT3-mediated EMT induction. Non-canonical TGF-β and EGF signaling induce the accumulation and upregulation of hypoxia-inducible factor 1α (HIF-1α) in the nucleus. During this process, the degradation of HIF-1α is blocked by the formation of STAT3-HIF-1α complex, and this is followed by induction of Twist-1 expression by binding of both STAT3 and HIF-1α to the Twist-1 and VEGF promoters [117,118,119,120]. Moreover, hypoxia-inducible ERO1α significantly correlates with poor prognosis and metastasis of HCC and enhances the metastatic ability and EMT via activation of S1PR1/STAT3/VEGF-A signaling [121]. 

Direct activation of STAT3 by the NPM-ALK chimeric protein, which plays a central pathogenic role in anaplastic large cell lymphoma (ALCL), promotes invasiveness of ALK-positive ALCL through induction of Twist-1 [122]. The kinase activity of IGFR induced by NPM-ALK-IGFR complex activates STAT3 and its downstream proteins (AKT, FKHR) that regulate cell survival [123]. Moreover, NPM–ALK suppresses STAT1-mediated interference with STAT3 signaling by promoting the ubiquitin-proteasome pathway-mediated degradation of STAT1 in a STAT3-dependent manner. Furthermore, the NPM-ALK complex eventually inhibits IFN-γ-mediated STAT1 pathway-induced suppression of tumor growth [124]. 

JAK/STAT3 signaling also can be induced by oncoproteins, including RPL34 [125], HOXB8 [126], CD146 [127], and Akirin2 [128]. This recently recognized regulation of new oncogenes involved in activating JAK/STAT3/VEGFA, NF-κB, and ERK signaling might provide crucial targets to block the cancer progression. 

Recently, it was reported that inhibition of the canonical TGF-β signaling due to loss of SMAD4 might be associated with cancer progression and drug resistance. Loss of SMAD4 induces constitutive activation of STAT3 in pancreatic cancer, which cooperates with non-canonical TGF-β signaling to promote cancer progression [129]. Screening via siRNAs identified STAT3 as a transcription factor, which is required for TGF-β and EGF-induced Snail production. TGF-β decreases the binding of PIAS3 to p-STAT3 through increasing the following: TGF-β-mediated formation of the PIAS3-SMAD3 complex and PIAS3-PAI-1 complex, resulting in Snail production via p-STAT3 [130,131,132]. Induction of stem cell factor (SCF)/c-KIT signaling by binding of SMAD2 to its promoter activates STAT3, which, in turn, induces TGF-β1 expression by binding to STB-2 region of TGF-β1 promoter via a positive feedback loop [133]. Moreover, TGF-β-induced histone acetyltransferase (GCN5) enhances EMT by activation of STAT3 and AKT and induces the transcriptional activity of TGF-β1 by interacting with R-Smads [134,135]. Intracellular signaling components such as receptors, non-receptor kinases, and other proteins instigate cancer-specific pleiotropic responses through JAK/STAT-mediated transcriptional regulation in cancer progression and aggressiveness.

## 5. Targeting the JAK2/STAT3 Signaling Pathway in EMT

There are multiple intrinsic regulators, which can prevent inappropriately sustained STAT3 activation (Figure 3A). Nitric oxide (NO) acts as a critical protector, and high secretion of NO, which is produced by NO synthase enzymes (NOS), blocks the induction of EMT by inhibiting TGF-β induced STAT3 activation in HCC [136]. Additionally, PPAR-γ and its agonists in combination with other drugs—such as type I interferons, gemcitabine, and COX-2 inhibitors—inhibit the survival of pancreatic cancer cells via IFN-β-induced activation of STAT-3, MAPK, and AKT [137,138].

AMPK has been implicated in the inhibition of tumor proliferation via suppression of STAT3 activation. AMPK suppresses tumor survival through inhibition of mTOR by increasing the expression of p-TSC2, a tumor suppressor gene [139], and leads to the inactivation of AKT and lipopolysaccharide (LPS)-induced IL-6/JAK2/STAT3 signaling in triple-negative breast cancer (TNBC) [140]. In addition, activation of the LKB1/AMPK pathway by metformin which is the first-line treatment for type 2 diabetes, suppresses IL-6-induced STAT3 expression and NF-κB activation in various cancers [141,142,143,144].

Tyrosine phosphatase is involved in the negative modulation of STAT3 activity. SHP-1 promotes the inactivation of JAK/STAT3 [145,146], and the STAT3/DNMT1/HDAC1 complex reduces SHP-1 expression by binding to the SHP-1 promoter [147]. Several other proteins negatively regulate JAK/STAT3 activation by inducing JAK degradation. SOCS-1 [148], SOCS-2 [149], and SOCS-3 [150] lead to proliferation and metastasis by suppression of JAK/STAT3 and p38 MAPK activation. In addition, SOCS-2 interacts with IGF1R and inhibits IGF1/IGF1R/JAK/STAT signaling [149]. 

Some negative regulators inhibit IL-6/JAK/STAT signaling by downregulating the expression of these proteins. ESE3/EHF, ETS transcription factor, inactivates IL6/JAK/STAT3 signaling through the transcriptional repression of IL-6 by direct binding to the IL-6 promoter [151]. Frequently methylated CHD5 also inhibits the invasion of RCC by suppression of oncogenes such as STAT3, epigenetic masters, and stem cell marker expression through direct binding to their promoters [152]. Moreover, apoptosis inducer, ING5, inhibits EGFR and IL-6 production, inactivates JAK/STAT3 and AKT pathways through inactivation of the Wnt/β-catenin pathway, and eventually prevents metastasis of lung cancer [153,154].

Alongside the transcriptional repression, critical negative regulation occurs via inhibition of IL-6/JAK/STAT3 activation. Mucosal barrier disruption by loss of CLDN-3 promotes CRC malignancy through induction of the Wnt/β-catenin signaling by activation of IL-6/JAK/STAT3 signaling. In contrast, the upregulation of CLDN-17 activates Tyk2/STAT3 signaling to promote malignancy in HCC [155,156]. In particular, the androgen receptor (AR) decreases macrophage recruitment by reduction of CCL2. Furthermore, AR loss enhances the EMT and invasion of cancer through induction of CCL2/CCR2/STAT3 axis [157]. However, STAT activation upregulates IL-6 and chemokine CCL2 expression, which significantly enhances the induction of EMT [158]. Moreover, high levels of GRIM-19 inhibit hypoxia-induced autophagy in CRC cells by repression of the activation of the STAT3/HIF-1α/VEGF axis under hypoxic conditions [159]. Furthermore, JAK2/STAT3 inactivation by ALK4 and FRK results in reduced cell survival and aggressiveness of different types of cancer. [160,161].

Recent studies have identified that other negative regulators inhibit JAK/STAT3 activation by interfering with the interaction between STAT3 and downstream targets or non-receptor tyrosine kinase. DNA JB4 inhibits Src phosphorylation by direct binding to the SH3 domain of c-Src and suppresses invasion of cancer by disrupting the interaction between Src and Src-downstream targets such as EGFR, FAK, and STAT3 [162]. RKIP inhibits the metastatic potential of cancer through the reduction of IL-6, Raf, or c-Src-mediated STAT3 activation by direct interaction with STAT3 [163,164]. NDRG2 suppresses the invasion abilities of cancer cells through the inhibition of EGF-induced STAT3/Snail signaling by inhibiting the binding of STAT3 to the Snail promoter [165]. Inhibition of IL-6-induced STAT3 activation by Dp44mT-induced NDRG2 attenuates TGF-β1-induced EMT in HCC [166].

Cytokines secreted by either cancer cells or surrounding stromal cells act as negative regulators. For instance, IL-32α and IL-32θ inhibit TGFβ-induced EMT, metastasis, and self-renewal in various cancer cells by inhibition of STAT3 activation [167,168]. Additionally, IL-37b enhances the survival of HCC patients and inhibits HCC progression by suppression of IL-6/STAT3 signaling and Gankyrin expression [169].

## 6. Link between JAK2/STAT3 Activation and the Transition of Cancer Stem Cells

Cancer stem cells have been proposed to explain cancer progression and therapeutic resistance. Multiple extracellular stimuli and intracellular signaling pathways have been implicated in the activation of the JAK/STAT signaling pathway or in the induction of cross-talk between various JAK/STAT pathways to promote the generation of cancer stem cells and acquisition of drug resistance [170,171]. Of the 28 cytokines and growth factors, OSM is the most potent inducer of CSC properties, and STAT3 activation is essential for OSM-induced EMT and generation of CSCs. Members of the IL-6 cytokine family—including IL-6, CNTF, CT-1, CLCF1, LIF, and OSM—significantly increase STAT3 activation and the subpopulation of CSCs. Additionally, OSM-induced JAK2/STAT3 activation increases Snail and HAS2 levels, which act as CD44 ligands and induce nuclear accumulation of p-SMAD3 by STAT3/SMAD3 complex, and subsequently, facilitate EMT and generation of CSCs [35,172]. The STAT3–SMAD3 complex suppresses TGF-β1-mediated antiproliferative signaling by inhibiting the interaction between SMAD3 and SMAD4 [173]. Moreover, induction and activation of Src by interaction with CD44 promote the survival of cancer cells in the matrix via the activation of integrin β1 in lipid rafts [174]. Additionally, the CD44/acetylated STAT3/p300 complex upregulates STAT3-mediated transcriptional activity of cyclin D1 [175]. After the formation of the CD44–STAT3 complex, STAT3 is activated and gets translocated into the nucleus. Several studies have demonstrated that NF-κB bound to STAT3 induces hTERT expression via binding to the hTERT promoter, which is significantly associated with aggressiveness and poor survival of patients with cancer [176,177]. Furthermore, the CD44/acetylated STAT3 complex increases the expression of stem cell markers (c-Myc, SOX2, and OCT4) [178]; SOX2 induces Slug expression via activation of STAT3/HIF-1α signaling [179]. Besides, the upregulation of IL-6 in CRC-derived mesenchymal stem cells (CC-MSCs) results in enhanced metastasis and survival of patients with CRC; this is attributed to the activation of PI3K/AKT via IL-6/JAK2/STAT3 signaling [180]. 

Osteopontin (OPN)/integrin complex upregulates the expression of CD44 variants. OPN binding to CD44 enhances metastasis and generation of CSC through activation of STAT3 [181,182,183]. Additionally, OPN as a tumor biomarker is induced through the TM4SF4/GSK3β/β-catenin axis, JAK2/STAT3, and FAK/STAT3 signaling, and is additionally regulated via formation of positive feedback autocrine loop that results in cells acquiring CSC-like properties [183]. 

JAK/STAT3 activation by positive regulators is involved in drug- or radiation-resistance. CXCR4-mediated STAT3 activation induces gefitinib-resistance and transition into stem-like cancer cells, which possess self-renewal capacity and radiation resistance [184]. Additionally, TM4SF5 expression induces EMT and drug resistance via the activation of FAK and promotion of self-renewal properties in HCC by repressing CD24, and interaction with CD44 to activate c-Src/STAT3/Twit-1/Bim1 signaling to induce sphere formation [185]. Moreover, drug resistance, acquired due to the prolonged trastuzumab treatment of GC, is associated with the remarkable increase in autocrine IL-6 production via IL-6-mediated STAT3 activation. Notably, both activation of JAK/STAT and Notch signaling by IL-6-induce Jagged-1, Hey1, and Hey2, leading to enhanced patient survival, accumulation of CSCs, and drug resistance in cancer [186]. Furthermore, Ras-induced Prx II enhances the self-renewal properties of cancer cells through the activation of VEGFR2/STAT3 signaling in HCC [187].

## 7. Role of microRNAs and JAK/STAT3 Activation in EMT and Transition into Cancer Stem Cells

### 7.1. MicroRNAs as Positive Regulators

In addition to JAK2/STAT3 signaling, which is mainly regulated by proteins (positive or negative modulators), the expression and functions of JAK/STAT are also controlled by other regulatory mechanisms. Notably, many studies have established a link between non-coding microRNAs (miRNAs) and JAK/STAT signaling. 

JAK/STAT3 signaling induces miRNAs to regulate the target genes. For instance, the loss of p53, IL-6, and LIF induces mTOR and STAT3 activation [188,189,190], leading to the increase in mir-21 levels, thereby inducing tumorigenesis [191], resistance to chemotherapy [192], induction of EMT by upregulation of CDK5 [193], and repression of PIAS3 [194]. Upregulation of miR-96/182/183 in BC patients due to autocrine/paracrine HGH-induced STAT3 and STAT5 activation enables tumor progression by BRMS1L [195]. The levels of miR-143 and miR-145 are increased in response to STAT3 activation. These miRNAs induce EMT by enabling the upregulation of EMT-TFs, Snail, and Slug through TGF-β, and downregulating epithelial cell markers and transcription factors, such as Creb, c-Fos, and Egr1 [196].

miRNAs are upregulated in response to STAT activation, and the JAK/STAT3, in turn, is often activated by miRNAs. miR-221/222 induces the EMT via activation of IL-6-dependent NF-κB and STAT3 through suppression of adiponectin receptor 1 (ADIPOR1) [197,198]. Additionally, miR-373 enhances CRC progression through the upregulation of cell cycle markers (CDK2, CDC2, and CCND2) and inflammation markers (NOS2, TGFβ1, CHI3L1, and TFPI) by induction of p-STAT3 [199]. Moreover, enhanced cell survival and tumorigenesis in BC are dependent on the activation of STAT3 by miR-30d-induced expression of KLF-11 [200]. 

Besides, increasing the stabilization of STAT3 through SOCS-1 or SOCS-2—both of which are downregulated by miR-155-5p or miR-194—promotes the spread of prostate cancer [201] and is associated with the poor survival of patients with oral squamous cell carcinoma (OSCC) [202].

### 7.2. MicroRNAs as Negative Regulators

miRNAs lead to the inactivation of the JAK/STAT signaling pathway through several different mechanisms. They act as negative regulators by suppressing the expression of IL-6/JAK/STAT3. miR-93 prevents the generation of CSCs and metastasis through the downregulation of multiple stem cell regulatory genes, including JAK1 and STAT3 [203]. Additionally, p53-induced miR-34a suppresses cancer metastasis by suppressing the expression of IL6R [204]. However, IL-6 or HIF-1A-induce STAT3 and InH3 expression, resulting in the induction of invasion and metastasis of CRC by repression of miR-34a [205]. Moreover, miR-216 inhibits the activation of JAK2/STAT3 signaling by reducing JAK2 expression [206], and miR-544 suppresses the expression of both Bcl6 and STAT3 by direct targeting [207]. Furthermore, miR-26a inhibits metastasis through the downregulation of IL-6 expression by reducing the expression of Lin-28 homolog B (LIN28B), HMGA1, and MITF as a direct target and through the restoration of LIN28B-reduced let7d expression [208,209]. Finally, miR-125a suppresses metastasis and cisplatin resistance in several types of cancer through repression of c-Myc, matrix metalloproteinase (MMP)-2, and MMP-9 by directly inhibiting STAT3 expression [210,211].

Additionally, miRNAs act as negative regulators by direct or indirect suppression of IL-6/JAK/STAT3 activation. miR-876-3p inhibits glioma progression, including EMT, by inhibiting JAK2/STAT3 activation via suppression of KIF20A expression [212]. GATA3-induced miR-30c decreases TWF1-induced IL-11/STAT3 activation through the downregulation of twinfilin 1 (TWF1) and vimentin (VIM) and correlates with poor survival and endocrine therapy resistance [213]. miR-126 inhibits the progression of osteosarcoma via targeting Zeb1 and inhibition of JAK1/STAT3 activation [214]. miR-7 inhibits SETDB1 expression and suppresses EMT and is involved in the generation of CSCs and dissemination of cancer via the inactivation of SETDB1-induced STAT3 signaling, which results in reduced c-Myc and Twist-1 levels due to the absence of STAT3 binding to the promoters of c-Myc and Twist-1 [215]. miR-33b inhibits EMT through repression of HMGA2 by deactivation of STAT3 [216]. miR-1207-5p, 505, and 148a also suppress cancer metastasis by suppressing the expression of corresponding genes via inhibition of AKT and STAT3 activation [217,218,219].

TCF-4/KIFC1-mediated HMGA1 induces the expression of STAT3, MMP2, and Twist-1 by binding to their promoters [220]. Additionally, HMGA2 directly induces the expression of IL-11, which promotes invasion in a STAT3 activation-dependent manner [221], but upregulation of the let-7 and miR-200 results in marked suppression of their target proteins, such as HMGA2 and Zab1. OSM-induced STAT3 activation restores the expression of HMGA2 and Zab1 through reducing let-7 and miR-200 levels. Moreover, STAT3 activation induces EMT by blocking let-7 maturation via the upregulation of Lin-28, which binds to STAT3 promoter [222].

The upregulation of thyroid hormone (T3) and its receptor (TR) in HCC suppresses miR-130b expression. However, increased miR-130b and miR-345 levels result in the upregulation of interferon regulatory factor 1 (IRF1) via T3/TR, which controls cell migration and invasion and represses the expression of p-mTOR, p-STAT3, p-AKT, and p-ERK1/2 [223,224]. The findings of many studies have highlighted the essential roles played by non-coding miRNAs in the regulation of the JAK/STAT signaling pathway (Figure 3B).

## 8. Conclusions

Along with its essential role in the progression of cancer, the IL-6/JAK/STAT3 signaling axis incorporates various unexpected components and miRNAs that contribute to JAK/STAT3 activation in cancer. Unpredicted functions of various components and miRNAs that have been identified as new positive or negative regulators of STAT3 may need further investigations, for us to be able to selectively target JAK/STAT3 signaling for cancer therapy. Thus, therapies that target these proteins or miRNAs, which are currently not thoroughly explored, may be implicated in suppressing invasion, dissemination, and transition into CSCs in the tumor microenvironment. Although small RNA-target based therapy, including miRNAs, is challenging due to the broad off-target effects, studies that investigate the use of small RNAs as medical interventions have expanded the preclinical and clinical applications. Recently, the FDA approved Onpattro (patisiran) for polyneuropathy in patients with hereditary amyloidosis; it is the first FDA approval for an siRNA drug. Thus, miRNAs or novel identified regulatory proteins may serve as potential targets in further clinical developments and represent significant conceptual advances in terms of repression of JAK/STAT3 activation in human diseases, including cancer, diabetes, inflammation, and neurodegenerative disorders. Thus, the development of therapeutic strategies based on the targeting of the JAK/STAT3 pathway is of utmost clinical relevance.

## Figures and Tables

**Figure 1 cells-09-00217-f001:**
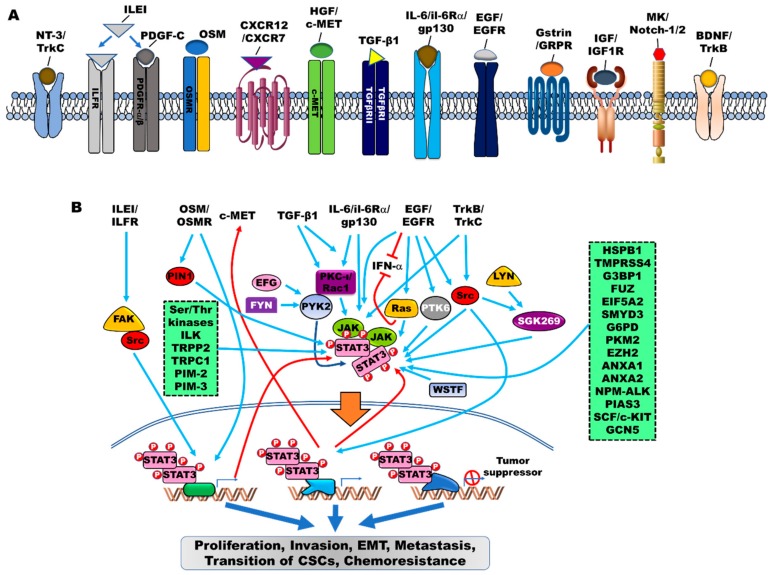
The contribution of signaling pathways that activate JAK/STAT3 signaling in cancer. Cytokines, growth factors, intracellular proteins, including non-receptor kinases (tyrosine or serine/threonine), can cooperate to induce the JAK/STAT3 signaling. (**A**) Various cytokines, peptide hormones, growth factors, and chemokines contribute to the activation of the JAK/STAT3 signaling to promote the progression of cancer. (**B**) The JAK/STAT3 signaling activated by tyrosine receptors and their cognate ligands, including neurotrophic receptors (TrkA, and TrkC), ILE/ILFR, PDGF-C/PDGFR, OSM/OSMR, CXCR12/CXCR7, HGF/c-MET, TGF-β/TGF receptors, IL-6/IL-6Rα/gp130, EGF/EGFR, Gastrin/GRPR, IGF/IGF1R, and Mk/Notch-1/2. Also, potential mechanisms by which tyrosine or serine/threonine kinases activate the JAK/STAT3 signaling through direct binding to JAK/STAT3 or indirect regulation of JAK/STAT3 activation. Once activated, phosphorylated and dimerized STAT3 enters the nucleus through importin-β1 and promotes the transcriptional expression of target genes to promote various cellular processes that are required for maintenance of survival in cancer.

**Figure 2 cells-09-00217-f002:**
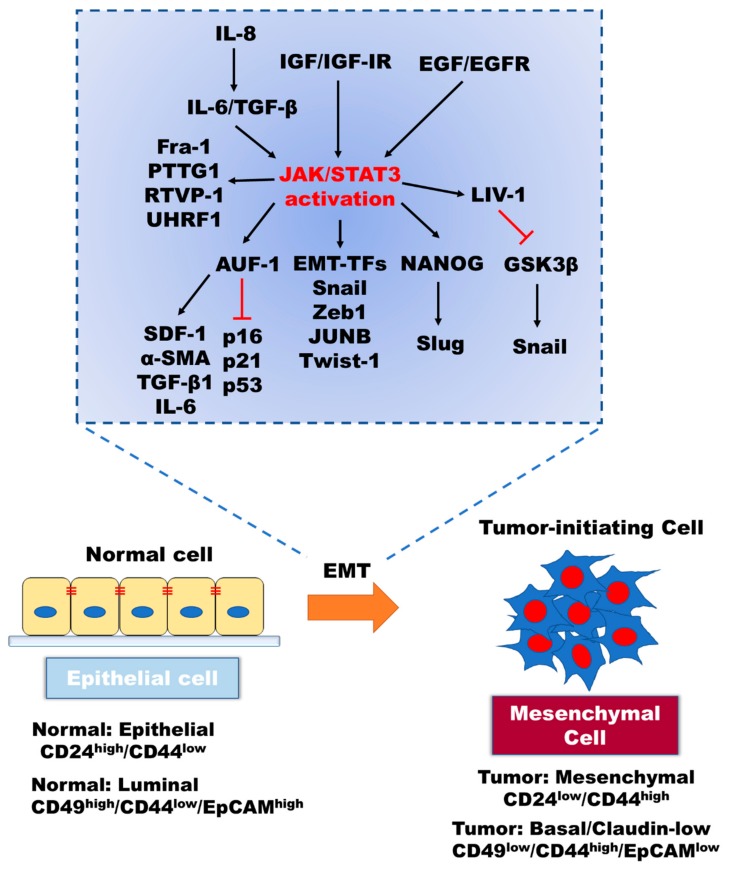
The contributions of intracellular signaling components to the inducing of EMT through activation of the JAK/STAT signaling. EMT program in cancer is associated with morphological and physiological changes of cancer, including the detachment of epithelial cells, enhanced invasiveness, tumor-initiating ability, and drug resistance. Enhanced EMT through upregulation of EMT-inducing transcription factors via the canonical and non-canonical JAK/STAT3 signaling leads to the downregulation of apoptotic signaling pathways, resistance to anticancer drugs and immunotherapy, induced cell proliferation, and CSC population.

**Figure 3 cells-09-00217-f003:**
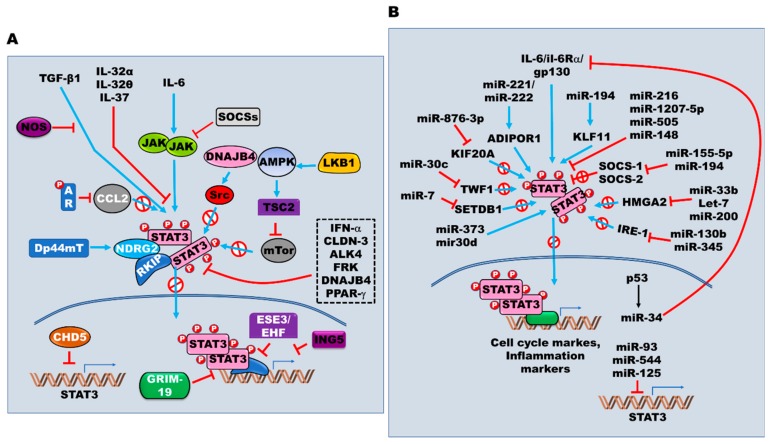
The regulation of the JAK/STAT3 signaling. (**A**) The multiple intrinsic proteins also involved in positive or negative regulation of JAK/STAT3 signaling with tyrosine receptors, their cognate ligands, and non-receptor kinases through ubiquitination, acetylation, regulation of STAT3 expression, regulation of STAT3 binding to the promoter of target genes, and direct physical interactions with STAT3. (**B****)** MicroRNAs-mediated regulation of the JAK/STAT3 activation. In addition to being regulated by multiple oncogenes and suppressors, the expression and activation of JAK/STAT3 are directly or indirectly controlled by microRNAs, which are endogenous small non-coding RNAs that function to the regulation of gene expression at transcriptional or post-transcriptional levels.

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
