# Peer review of "Role of JAK/STAT3 Signaling in the Regulation of Metastasis, the Transition of Cancer Stem Cells, and Chemoresistance of Cancer by Epithelial–Mesenchymal Transition"

_cells, 2020, doi:10.3390/cells9010217_

Round 1

Reviewer 1 Report

I think the review is an important and interesting topic.  The article and figures do a good job of outlining the multiple ways JAK/STAT signaling can be regulated and alternatively act as regulators.

I don't think the current state of the English used in this paper allows me to make a fair review at this time.   In particular, article usage (a, an, the)  is problematic with inappropriate usage and omission being noted. Verb tenses are not always accurate and change within sentences.

Figures are very nice and clear.  However, they should be modified to reduce white space and increase font size as much as possible.

Author Response

I think the review is an important and interesting topic.  The article and figures do a good job of outlining the multiple ways JAK/STAT signaling can be regulated and alternatively act as regulators.

I don't think the current state of the English used in this paper allows me to make a fair review at this time.   In particular, article usage (a, an, the)  is problematic with inappropriate usage and omission being noted. Verb tenses are not always accurate and change within sentences.

We apologize for the grammatical errors and have edited our manuscript again. We have corrected and modified the sentence by the English language editing service, as suggested by the Reviewer.

Figures are very nice and clear.  However, they should be modified to reduce white space and increase font size as much as possible.

We thank the Reviewer for this comment. As suggested by the Reviewer, we modified figures to reduce white space and increase the font size as much as possible.

Reviewer 2 Report

The present review is well written and has some scientific interest for the community summarizing the roles of STAT3 in cancer cell biology.

However, some major changes are required in order to achieve publication priority.

1) The authors should add a paragraph describing the interactions between Interferon alpha/EGFR and STAT3 citing relevant manuscripts such as:

Caraglia et al. Cell Death Differ. 2003 Feb;10(2):218-29. Boccellino et al. J Cell Physiol. 2004 Oct;201(1):71-83

This response is based upon n increased biosynthesis of EGFR and the potentiation of Erk1/2 and STAT3 function that counteract the apoptotic and inhibitory activity of Interferon alpha.

2) The authors should cite the cross talk between PPARgamma and STAT3 citing the following relevant papers: Vitale et al. Biotechnol Adv. 2012 Jan-Feb;30(1):169-84; Dicitore et al. Curr Cancer Drug Targets. 2013 May;13(4):460-71

3) In one of the figures the interaction between EGFR/Interferon alha and STAT3 and PPARgamma and STAT3 should be added.

Author Response

The present review is well written and has some scientific interest for the community summarizing the roles of STAT3 in cancer cell biology.

However, some major changes are required in order to achieve publication priority.

 1) The authors should add a paragraph describing the interactions between Interferon alpha/EGFR and STAT3 citing relevant manuscripts such as:

Caraglia et al. Cell Death Differ. 2003 Feb;10(2):218-29. Boccellino et al. J Cell Physiol. 2004 Oct;201(1):71-83

This response is based upon n increased biosynthesis of EGFR and the potentiation of Erk1/2 and STAT3 function that counteract the apoptotic and inhibitory activity of Interferon alpha.

As suggested by the Reviewer, we added a paragraph describing the interactions between Interferon alpha/EGFR and STAT3 and cited (line 171-174).

2) The authors should cite the cross talk between PPARgamma and STAT3 citing the following relevant papers: Vitale et al. Biotechnol Adv. 2012 Jan-Feb;30(1):169-84; Dicitore et al. Curr Cancer Drug Targets. 2013 May;13(4):460-71

As suggested by the Reviewer, we added a paragraph cross-talk between PPARgamma and STAT3 and cited (line 281-284).

3) In one of the figures the interaction between EGFR/Interferon alha and STAT3 and PPARgamma and STAT3 should be added.

As suggested by the Reviewer, we added the interaction between EGFR/Interferon alpha and STAT3 and PPARgamma and STAT3 in figure 1B, 2, and 3A.

Round 2

Reviewer 1 Report

The review is well written and the authors have done a great job of responding to reviewers comments. Overall it is a great synthesis of many different contributions into the review.

I think that the service has done a reasonable job of clarifying the English in this article.  You may want to ask that they consider the overuse of words like furthermore, moreover, notably, and additionally as it interrupts the flow of some sections.

Two other minor comments.

Line 24.  Played should be plays

Line 206 remove repeated words

Reviewer 2 Report

The authors have adequately addressed the referees concerns.